# KPC-2 allelic variants in *Klebsiella pneumoniae* isolates resistant to ceftazidime-avibactam from Argentina: *bla*$_{KPC-80}$, *bla*$_{KPC-81}$, *bla*$_{KPC-96}$ and *bla*$_{KPC-97}$

María Belén Sanz,[1] Fernando Pasteran,[1] Juan Manuel de Mendieta,[1] Florencia Brunetti,[2,3] Ezequiel Albornoz,[1] Melina Rapoport,[1] Celeste Lucero,[1] Laura Errecalde,[4] Maria Rosa Nuñez,[5] Renata Monge,[6] Magdalena Pennini,[7] Pablo Power,[2,3] Alejandra Corso,[1] Sonia A. Gomez[1,3]

**ABSTRACT**  Ceftazidime-avibactam (CZA) therapy has significantly improved survival rates for patients infected by carbapenem-resistant bacteria, including KPC producers. However, resistance to CZA is a growing concern, attributed to multiple mechanisms. In this study, we characterized four clinical CZA-resistant *Klebsiella pneumoniae* isolates obtained between July 2019 and December 2020. These isolates expressed novel allelic variants of *bla*$_{KPC-2}$ resulting from changes in hotspots of the mature protein, particularly in loops surrounding the active site of KPC. Notably, KPC-80 had an K269_D270insPNK mutation near the Lys270-loop, KPC-81 had a del_I173 mutation within the Ω-loop, KPC-96 showed a Y241N substitution within the Val240-loop and KPC-97 had an V277_I278insNSEAV mutation within the Lys270-loop. Three of the four isolates exhibited low-level resistance to imipenem (4 µg/mL), while all remained susceptible to mero-penem. Avibactam and relebactam effectively restored carbapenem susceptibility in resistant isolates. Cloning mutant *bla*$_{KPC}$ genes into pMBLe increased imipenem MICs in recipient *Escherichia coli* TOP10 for *bla*$_{KPC-80}$, *bla*$_{KPC-96}$, and *bla*$_{KPC-97}$ by two dilutions; again, these MICs were restored by avibactam and relebactam. Frameshift mutations disrupted *omp*K35 in three isolates. Additional resistance genes, including *bla*$_{TEM-1}$, *bla*$_{OXA-18}$ and *bla*$_{OXA-1}$, were also identified. Interestingly, three isolates belonged to clonal complex 11 (ST258 and ST11) and one to ST629. This study highlights the emergence of CZA resistance including unique allelic variants of *bla*$_{KPC-2}$ and impermea-bility. Comprehensive epidemiological surveillance and in-depth molecular studies are imperative for understanding and monitoring these complex resistance mechanisms, crucial for effective antimicrobial treatment strategies.

**IMPORTANCE** The emergence of ceftazidime-avibactam (CZA) resistance poses a significant threat to the efficacy of this life-saving therapy against carbapenem-resist-ant bacteria, particularly *Klebsiella pneumoniae*-producing KPC enzymes. This study investigates four clinical isolates exhibiting resistance to CZA, revealing novel allelic variants of the key resistance gene, *bla*$_{KPC-2}$. The mutations identified in hotspots surrounding the active site of KPC, such as K269_D270insPNK, del_I173, Y241N and V277_I278insNSEAV, prove the adaptability of these pathogens. Intriguingly, low-level resistance to imipenem and disruptions in porin genes were observed, emphasizing the complexity of the resistance mechanisms. Interestingly, three of four isolates belonged to clonal complex 11. This research not only sheds light on the clinical significance of CZA resistance but also shows the urgency for comprehensive surveillance and molecular studies to inform effective antimicrobial treatment strategies in the face of evolving bacterial resistance.

Address correspondence to Sonia A. Gomez, sgomez@anlis.gob.ar.

The authors declare no conflict of interest.

See the funding table on p. 11.

**KEYWORDS** ceftazidime/avibactam, KPC, *Klebsiella pneumoniae*, allelic variant

Carbapenemase-producing Enterobacterales (CPE) pose a significant and high-priority microbial threat. These microorganisms are particularly worrisome because they not only display resistance to almost all β-lactam antibiotics but frequently exhibit resistance to other classes of antibiotics as well (1). Ceftazidime-avibactam is a combination of a β-lactam antibiotic and a β-lactamase inhibitor with the potential to treat severe infections caused by carbapenem-resistant organisms (2). Avibactam restores the activity of ceftazidime by inhibiting Ambler Class A, Class C, and some Class D β-lactamases, including KPC and OXA-48 carbapenemases (3, 4). It is indicated for the treatment of complicated urinary tract infections, including pyelonephritis, and hospital-acquired pneumonia, including ventilator-associated pneumonia, in adults and pediatric patients aged 3 months and older (5). In some countries, including Argentina, it is prescribed for treating infections caused by aerobic Gram-negative bacteria with limited treatment options. Ceftazidime-avibactam therapy has shown significant clinical success and survival rate for patients infected by target bacteria. A delayed initiation of treatment with ceftazidime-avibactam has been associated with worse clinical and microbiological outcomes (2).

Recently, resistance to ceftazidime-avibactam has been reported, associated with mutations near the active site of KPC (e.g., omega-loop), alterations in porins, the presence of extended-spectrum beta-lactamases like PER-2 or mutations in PBP3 that have a specific affinity for ceftazidime, among others (6).

The general structure of the KPC protein, like other Class A β-lactamases, consists of subdomains composed of alpha-helices and beta-sheets, which are conserved and contribute to the catalytic activity of the enzyme (7). It has been demonstrated *in vitro* that different variations in the amino acid sequence of the protein lead to distinct profiles of resistance to β-lactam antibiotics, including ceftazidime-avibactam resistance (8). In addition, the occurrence of ceftazidime-avibactam-resistant isolates expressing KPC with mutations in the loops surrounding the active site has been described in clinical practice like KPC-41, a KPC-3 variant (9), or KPC-28, a KPC-2 variant (10). These mutations, often insertions or deletions, highlight KPC's evolutionary capacity. Particularly in the omega-loop, such mutations can modify salt bridges, enhancing loop flexibility and altering the substrate spectrum (11).

In Argentina, KPC-2 was first reported in 2008 and disseminated throughout the country when *Klebsiella pneumoniae* ST258 entered in 2010, becoming endemic thereafter (12, 13). Until 2019, the only allelic variant circulating in Argentina was KPC-2, until the first report of KPC-3 in a *K. pneumoniae* ST307 (14). Consequently, the use of new drug combinations, such as ceftazidime-avibactam, for treating infections caused by carbapenemase-producing Gram-negative bacteria gained importance. In this context, four *Klebsiella pneumoniae* clinical isolates obtained between July 2019 and December 2020 with resistance to ceftazidime-avibactam were referred to the National and Regional Reference Laboratory for Antimicrobial Resistance (NRRLAR) for further study. Here, we aimed to analyze the cause of ceftazidime-avibactam resistance through epidemiological, phenotypic and molecular studies of the clinical isolates.

## MATERIALS AND METHODS

### Bacterial isolates

Four ceftazidime-avibactam-resistant bacterial clinical isolates, M25399, M25752, M25197 and M25923, were sent to the NRRLAR for phenotypic confirmation of emerging ceftazidime-avibactam resistance and further molecular studies (15). The bacterial species were confirmed using matrix-assisted laser desorption/ionization time-of-flight mass spectrometry.

## Phenotypic characterization

The isolates underwent a comprehensive phenotypic analysis using various methods. These included the synergy test involving a 10-µg carbapenem and 30-µg aztreonam disk placed ca. 20 mm and 10 mm, respectively, to a disk containing 300 µg amino-phenyl boronic acid. In addition, 10-µg carbapenem and 10/4-µg ceftazidime-avibactam disks were placed to ca. 20 mm and 10 mm, respectively, to a disk containing 750 µg EDTA (16). Colorimetric assays such as Blue Carba and Carba NP Direct (17, 18), the Triton Hodge Test microbiological assay (19), immunochromatographic assays (RESIST-3, Coris BioConcept, Gembloux, Belgium) were performed in order to characterize the isolates. Evaluation of enzymatic activity was performed using modified carbapenem inactivation method (mCIM/eCIM) tests (18). The minimum inhibitory concentration (MIC) was determined by agar dilution (18).

To ensure the consistency and accuracy of susceptibility results, the guidelines established by the Clinical and Laboratory Standards Institute were followed (18). Exceptions were the breakpoints for colistin, fosfomycin, tigecycline and ceftazidime-avibactam for which the European Committee on Antimicrobial Susceptibility Testing Breakpoint tables for interpretation of MICs and zone diameters (https://www.eucast.org/clinical_breakpoints) were used.

## Molecular biology analysis

The $bla_{KPC}$ genes were initially confirmed using a multiplex PCR set up at the NRRLAR to detect epidemiologically relevant carbapenemases. The multiplex was set up for five epidemiologically relevant carbapenemases: $bla_{OXA-48-like}$, $bla_{KPC}$, $bla_{NDM}$, $bla_{IMP}$, and $bla_{VIM}$ (20). PCR for $bla_{KPC}$ was performed to confirm the nucleotide sequence by Sanger technology ABI PRISM 3100 (Applied Biosystems). GeneXpert-Xpert Carba-R (XCR) (Cepheid) was additionally used to confirm $bla_{KPC}$ detection. The amino acid alignment and structural visualization were performed using ESPript 3.0 and RCSB PDB, respectively (21, 22). The integrity of porin genes $ompK35$ and $ompK36$ was preliminarily detected by PCR and agarose gel electrophoresis using a K. pneumoniae ATCC 13883 harboring wild-type $ompK35$ and $ompK36$ as a control strain A (23).

## Biparental conjugation

Biparental conjugation assay of ceftazidime-avibactam-resistant isolates was performed on solid medium using Escherichia coli J53 (azide resistant) as acceptor strain. Transconjugant strains were identified by conventional biochemical methods and selected on Tryptic Soy Agar plates containing 200 µg/mL of azide and 50 µg/mL of ampicillin or 10 µg/mL of ceftazidime. The horizontal transfer of the putative ceftazidime-avibactam-resistant determinant was evaluated by disc diffusion and PCR.

## Cloning of $bla_{KPC}$ allelic variants and susceptibility testing of recombinant clones

The complete $bla_{KPC}$ genes were amplified from whole DNA of the corresponding clinical strain by PCR using primers designed to introduce the NdeI and EcoRI restriction sites: KPC-F-NdeI (5′CATATGTCACTGTATCGCC3′) and KPC-R-EcoRI (5′GAATTCTTACTGCCCGTT3′). A proof-reading Pfu polymerase (Thermo Scientific, USA) was used in PCR reactions to avoid errors in the $bla$ gene amplification. Amplified and purified amplicons were cloned into a pGEM-T Easy Vector (Promega, USA), and the resulting constructions were transformed into chemically competent E. coli TOP10F′ cells. The presence of the inserts and restriction sites were verified by DNA sequencing (Macrogen, South Korea). For subsequent cloning, the amplicons were digested from the original construction, and the released fragments were purified and then ligated in the NdeI and EcoRI sites of a pMBLe vector. Ligation mixtures were transformed in chemically competent E. coli TOP10F′ cells, and recombinant clones were selected in lysogeny broth (LB) agar supplemented with

20 µg/mL gentamicin. Recombinant plasmids of the selected clones were extracted and sequenced to verify the identity of *bla* genes and their proper insertion.

The MICs were determined for the *E. coli* TOP10 clones carrying pMBLe with the variants $bla_{KPC-80}$, $bla_{KPC-81}$, $bla_{KPC-96}$ and $bla_{KPC-97}$ along with the *E. coli* TOP10 wild type, *E. coli* TOP10 transformed with closed pMBLe, and *E. coli* TOP10 transformed with pMBLe with $bla_{KPC-2}$.

## Whole genome sequencing and bioinformatic analysis

DNA was extracted using the QIAamp DNA Mini kit (Qiagen) following the manufacturer's instructions. The DNA concentration was determined using the Qubit 2.0 fluorometer (Thermo Fisher Scientific). The library was generated using the Nextera XT DNA Library Preparation Kit following the manufacturer's instructions. Whole genome sequencing was performed using the Illumina MiSeq platform to generate paired-end reads of 250 bp at the National Center of Genomics and Bioinformatics, ANLIS "Dr. Carlos G. Malbrán." Quality assessment of the reads was performed using FASTQC (V.0.11.5) (24), and Kraken2 (V.2.0.7-beta) was used to confirm the species (25). The reads were *de novo* assembled using SPAdes (3.13.0), and the assembly quality was evaluated using QUAST (V.5.0.2) (26). Genome annotation was performed using Prokka (V1.14.6) (27). The sequence type (ST) for each genome was also determined using ARIBA MLSTdb (V.2.14.6) (28). The resistance genes were determined using ARIBA Resfinder (V.2.14.6) (29), and the corresponding gene alleles were confirmed through *in silico* assemblies and AMRFinder-Plus (V.3.8.4) (30). The analysis of the genetic environment of $bla_{KPC}$ was performed using TetTyper (31) and the capsular type with Kaptive 2.0 (32). The sequencing quality was optimal, and the identification of the corresponding bacterial species was consistent with MaldiTOF results (Table S1). Likewise, the assembly quality provided genome sizes and %GC content appropriate for the species under study (Table S2).

## RESULTS

### Epidemiological and phenotypic analyses

The epidemiological data and phenotypic results are detailed in Table 1. Briefly, the isolates were received at the NRRLAR between November 2019 and December 2020 from four institutions. Among these, one isolate was recovered from a patient in Neuquén Province Hospital, while the remaining three were from different hospitals in Buenos Aires City (CABA) with no apparent epidemiological connection. The average age of the patients was 58.5 years, two females and two males. Three isolates were obtained from infection sites and one from a screening procedure. Of note, the patients infected or colonized with *K. pneumoniae* M25752 and *K. pneumoniae* M25923 had previously undergone treatment with ceftazidime-avibactam (Table 1).

The presence of a carbapenemase activity was confirmed by Triton-Hodge microbiological assay and GeneXpert-Xpert Carba-R, with all four isolates testing positive. None of the variants was detected by the colorimetric tests Blue Carba Test or CarbaNP. Additional tests, such as double disk synergy, mCIM, and immunochromatography, yielded variable results for the different isolates (Table 1).

All isolates demonstrated resistance to ceftazidime, ceftazidime-avibactam, cefotaxime, and aztreonam. All isolates had low-level resistance to imipenem, except for KPN-$bla_{KPC-81}$, which showed susceptibility. Regarding meropenem, all isolates were susceptible, except for KPN-$bla_{KPC-97}$ with an intermediate MIC result. Avibactam or relebactam restored carbapenem and aztreonam susceptibility, as shown in Table 2. Avibactam failed to restore ceftazidime susceptibility, whereas on other cephalosporins tested, the inhibitory effects of avibactam varied. Only fosfomycin was uniformly susceptible on all the isolates. Only KPN-$bla_{KPC-80}$ was susceptible to colistin, while all strains were resistant to tigecycline, ciprofloxacin and trimethoprim-sulfamethoxazole (Table 2).

**TABLE 1** Epidemiological data and phenotypic test results[a]

| Parameter | M25399 | M25752 | M25197 | M25923 |
|---|---|---|---|---|
| KPC allele | $bla_{KPC-80}$ | $bla_{KPC-81}$ | $bla_{KPC-96}$ | $bla_{KPC-97}$ |
| Referring institution | A | B | C | D |
| Province | CABA | NEUQUEN | CABA | CABA |
| Isolation date | 11 November 2019 | 10 July 2020 | 26 July 2019 | nd |
| Received NRRLAR date | 22 November 2019 | 17 July 2020 | 12 August 2019 | 11 December 2020 |
| Gender | F | M | F | M |
| Age | 64 | 42 | 61 | 67 |
| Prior antimicrobial treatment | Meropenem plus tigecycline | Ceftazidime-avibactam plus linezolid | Meropenem | Ceftazidime-avibactam |
| Sample | Abdominal fluid | Urine | Urine (by catheter) | Rectal swab |
| Phenotypic and molecular screening of carbapenemase production | | | | |
| Immunochromatography[b] | Neg | Pos(KPC) | Neg | wPos(KPC) |
| Carbapenemase colorimetric methods[c] | Neg | Neg | Neg | Neg |
| mCIM | Pos | Pos | Neg | Neg |
| THT | Pos | Pos | wPos | Pos |
| Double disk synergy test (i) Sinergy: IMI-APB-FOX | Pos | Pos | Neg | Pos |
| (ii) Sinergy: CAZ-AMC-CTX | Pos | Pos | Pos | Pos |
| GeneXpert-Xpert Carba-R and PCR | KPC | KPC | KPC | KPC |
| Conjugation | Pos | Neg | Neg | Neg |

[a]CABA, Ciudad Autónoma de Buenos Aires; F, female; M, male; Neg, negative; Pos, positive; wPos, weak positive; mCIM, modified carbapenem inactivation method; THT, Triton-Hodge test; IMI, imipenem; APB, phenyl boronic acid; FOX, cefoxitin; CAZ, ceftazidime; AMC, amoxicillin/clavulanate; CTX, cefotaxime; nd, not determined.
[b]Results were read after 15 minutes (per insert) and 60 minutes (this paper).
[c]Blue Carba Test and Carba NP-Direct.

**TABLE 2** Susceptibility testing of *K. pneumoniae* clinical isolates[a]

| Drug | MIC (µg/mL) | | | |
|---|---|---|---|---|
| | M25399- $bla_{KPC-80}$ K269_D270insPNK | M25752- $bla_{KPC-81}$ del_I173 | M25197- $bla_{KPC-96}$ Y241N | M25923- $bla_{KPC-97}$ V277_I278insNSEAV |
| Imipenem | 4 | 0.5 | 4 | 4 |
| Imipenem + avibactam | 0.12 | 0.12 | 0.008 | 0.5 |
| Imipenem + relebactam | 0.25 | 0.12 | 0.12 | 1 |
| Meropenem | 0.5 | 0.25 | 0.5 | 2 |
| Meropenem + avibactam | 0.032 | 0.12 | 0.008 | 0.125 |
| Ceftazidime | 128 | >256 | 32 | >256 |
| Ceftazidime + avibactam | 128 | >256 | 16 | >256 |
| Cefotaxime | 16 | 4 | 8 | 16 |
| Cefotaxime + avibactam | 0.5 | 0.5 | 0.5 | 2 |
| Ceftaroline | 16 | 32 | 32 | 128 |
| Ceftaroline + avibactam | 4 | 2 | 0.25 | 8 |
| Aztreonam | 32 | 16 | 128 | 64 |
| Aztreonam + avibactam | 0.25 | 0.5 | 0.125 | 1 |
| Colistin | ≤1 | 8 | >8 | 8 |
| Amikacin | 16 | 16 | 4 | 32 |
| Gentamicin | ≥16 | ≥16 | ≥16 | 4 |
| Ciprofloxacin | ≥4 | ≥4 | ≥4 | ≥4 |
| SXT | ≥4 | ≥4 | ≥4 | ≥4 |
| Fosfomycin | 8 | 8 | 8 | 16 |
| Tigecycline | 2 | 4 | 4 | 2 |

[a]SXT: trimethoprim/sulfamethoxazole.

## Molecular analysis of *K. pneumoniae* isolates harboring *bla*<sub>KPC</sub> variants

The nucleotide sequences of *bla*$_{KPC}$ revealed that the four isolates harbored novel allelic variants derived from *bla*$_{KPC-2}$, designated as *bla*$_{KPC-80}$, *bla*$_{KPC-81}$, *bla*$_{KPC-96}$ and *bla*$_{KPC-97}$. The nucleotide alignment of the different variants showed that *bla*$_{KPC-80}$ differs from *bla*$_{KPC-2}$ by a 9-bp insertion/duplication (CCTAACAAG, nucleotide position 796 to 804), resulting in the insertion of three amino acids (ProAsnLys) between Lys269 and Asp270 (K269_D270insPNK) near the Lys270-loop. *bla*$_{KPC-81}$ differs from *bla*$_{KPC-2}$ by a 3-bp deletion (ATC, nucleotide position 514 to 516), leading to the deletion of one amino acid at position Ile173 (del_I173) within the Ω-loop. *bla*$_{KPC-96}$ differs from *bla*$_{KPC-2}$ by a single-nucleotide point mutation (TxA, nucleotide position 718), resulting in the substitution Tyr241Asn (Y241N) within the Val240-loop. *bla*$_{KPC-97}$ differs from *bla*$_{KPC-2}$ by a 15-bp insertion (AACAGCGAGGCCGTC, nucleotide position 829 to 844), leading to the insertion of five amino acids between Val277 and Ile278 from the KPC-2 numbering scheme (V277_I278insNSEAV), within the Lys270-loop (Fig. 1). The amino acid alignment showing the mutated region of the new allelic variants with respect to *bla*$_{KPC-2}$ is shown in Fig. 1.

With the aim to determine whether the *bla*$_{KPC}$ allelic variants were located on mobile or conjugative plasmids, we performed biparental conjugation assays. As a result, a transconjugant was obtained only for M25399-KPN-*bla*$_{KPC-80}$ when selecting with azide and ceftazidime. In addition, the transconjugant strain was resistant to ceftazidime-avibactam with a MIC value of 8 dilutions higher than that of the wild-type *E. coli* J53 (32 µg/mL vs. 0.125 µg/mL, respectively).

## Phenotypic characterization of cloned KPC allelic variants

In order to study the phenotypic effect of the KPC enzymes without the genetic background of the clinical strains, *bla*$_{KPC}$ genes were cloned and transformed in fully susceptible competent cells (*E. coli* TOP10) (Table 3). The presence of pMBLe carrying *bla*$_{KPC-80}$ (K269_D270insPNK), *bla*$_{KPC-96}$ (Y241N) and *bla*$_{KPC-97}$ (V277_I278insNSEAV) resulted in an increase in the MICs of imipenem for the recipient *E. coli* TOP10, raising it by two dilutions and placing it in the intermediate category (Table 3). Conversely, there were no observed changes in the MIC when pMBLe carried *bla*$_{KPC-81}$ (del_I173). The influence on meropenem MICs was minimal for all four constructions. As a control, pMBLe carrying *bla*$_{KPC-2}$ was used, leading to an increase in MICs for both carbapenems. When imipenem was evaluated in the presence of both avibactam and relebactam, a significant two-dilution reduction in imipenem MIC was observed for the pMBLe constructs with *bla*$_{KPC-80}$, *bla*$_{KPC-96}$ and *bla*$_{KPC-97}$, lowering the values back to the TOP10 baseline.

In the case of cephalosporins and monobactams, the MICs for cefotaxime, ceftazidime, ceftaroline and aztreonam increased by a factor of 4 to 9 when compared with *E. coli* TOP10 transformed with the closed pMBLe. Interestingly, the MICs for these new variants in the case of ceftazidime were significantly higher than those observed in the *bla*$_{KPC-2}$ transformant. Avibactam had the capacity to restore the MICs of all cephalosporins and monobactams to the baseline TOP10 values, except for ceftazidime, where the MIC reduction was more modest, ranging between one and two dilutions. The weakest inhibitory effect of avibactam on ceftazidime MIC was observed for *bla*$_{KPC-96}$.

## Whole genome analysis of the clinical isolates harboring *bla*<sub>KPC</sub> allelic variants

Upon further analysis of these results, bioinformatic tools such as Resfinder-ARIBA identified KPC variants by reporting nucleotide identity percentages lower than 100% compared with the sequences in the database (Table 4). The genetic environment of *bla*$_{KPC-80}$ (K269_D270insPNK), *bla*$_{KPC-81}$ (del_I173) and *bla*$_{KPC-97}$ (V277_I278insNSEAV) was identified as Tn*4401a*, while *bla*$_{KPC-96}$ (Y241N) was found within a Tn*4401b-1* element.

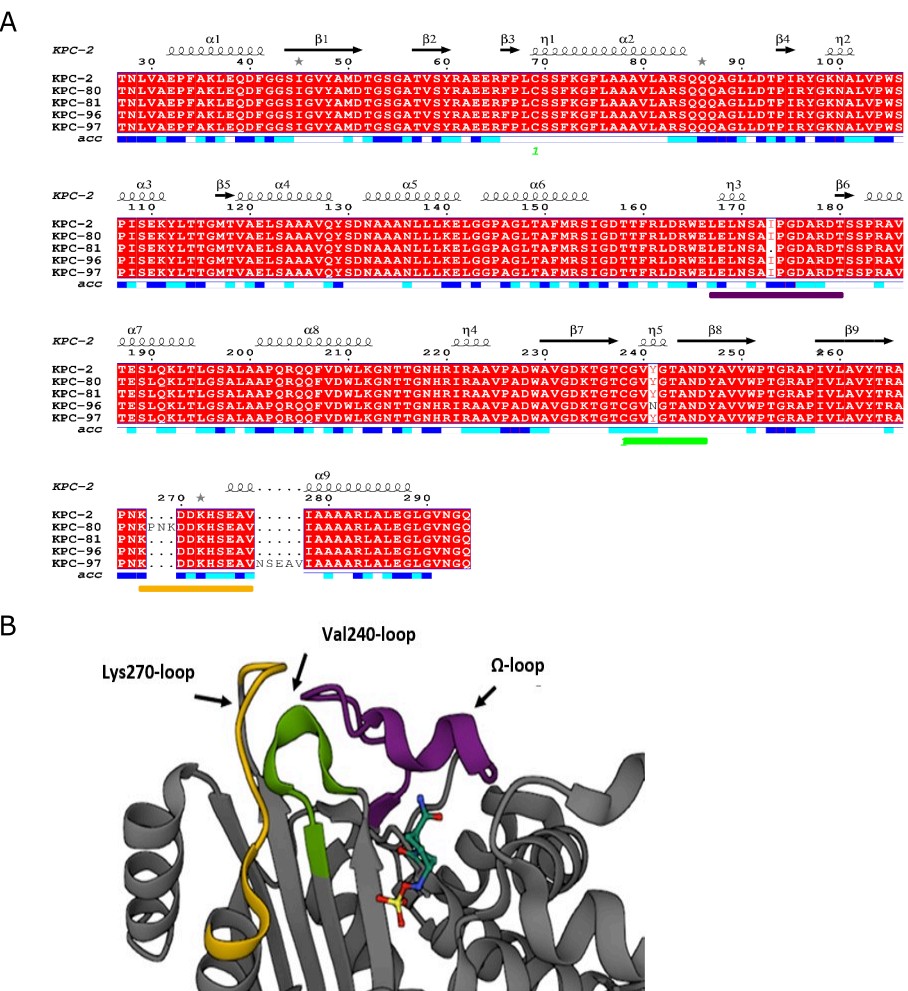

**FIG 1** Amino acid alignment of KPC-2 with the novel allelic variants. (A) The amino acid alignment was obtained with ESPript program. The image shows the α-helices and η-helices marked with loops; the β-strands are indicated with black arrows. The color-labeled lines indicate the position of the omega-loops (violet), Val240-loop (green), and Lys270-loop (yellow). (B) Fragment of a representative view of the KPC-2 fold (PDB 4ZBE, https://www.rcsb.org/3d-view/4ZBE) complexed with avibactam (bars) with colored and labeled loops Ω-loop, Val240, and Lys270 near the active site (33).

In order to determine additional factors that could contribute with ceftazidime-avibactam resistance phenotype, the integrity of the porin genes *omp*K35 and *omp*K36 was analyzed (Table 4). The strains carried mutations caused by insertions that rendered reading frame shifts and, as a consequence, unfunctional OmpK35 proteins in all genomes, except in M25399 (*bla*KPC-80) (Table 4). *omp*K36, *omp*K37 and *omp*S maintained the wild-type sequence in all genomes. Additionally, the four strains harbored the wild-type PBP3 encoding *fts1* gene. Furthermore, none of the isolates carried *bla*PER or AmpC genes (Table 4), also known to contribute with ceftazidime-avibactam resistance (6). The detailed accompanying resistome can be found in table S3. Briefly, all strains harbored *bla*SHV-11 and *fosA*, which is consistent with the *K. pneumoniae* species. They also carried *oqxA* and *oqxB*, *sul1*, and D87N mutation in DNA-gyrase. Additionally, the *qacEdelta1* gene responsible for resistance to quaternary ammonium biocides was present in all four isolates.

Three previously described ST of *K. pneumoniae*, ST258 (*n* = 2), ST11 (*n* = 1) and ST629 (*n* = 1) were identified. Three of these isolates belonged to the hyper epidemic clonal complex 11 (CC11).

**TABLE 3** Susceptibility testing of *E. coli* TOP10 cells transformed with mutant-KPC genes[a]

| E. coli TOP10 | MIC (µg/mL) | | | | | | | | | | | | |
|---|---|---|---|---|---|---|---|---|---|---|---|---|---|
| | Meropenem | | Imipenem | | | Ceftazidime | | Cefotaxime | | Ceftaroline | | Aztreonam | |
| | – | AVI | – | AVI | REL | – | AVI | – | AVI | – | AVI | – | AVI |
| – | 0.06 | 0.06 | 0.5 | 0.5 | 0.5 | 0.25 | 0.5 | 0.125 | 0.125 | 0.125 | 0.125 | 0.015 | 0.015 |
| pMBLe | 0.125 | 0.06 | 1 | 0.5 | 0.5 | 0.25 | 0.25 | 0.125 | 0.125 | 0.125 | 0.125 | 0.125 | 0.125 |
| pMBLe + *bla*KPC-2 | 0.5 | 0.125 | 16 | 0.5 | 0.5 | 2 | 0.5 | 8 | 0.125 | 64 | 0.25 | 16 | 0.125 |
| pMBLe + *bla*KPC-80 K269_D270insPNK | 0.125 | 0.125 | 2 | 0.5 | 0.5 | 16 | 4 | 4 | 0.25 | 8 | 0.5 | 1 | 0.125 |
| pMBLe + *bla*KPC-81 del_I173 | 0.125 | 0.125 | 0.5 | 0.5 | 0.5 | 16 | 4 | 4 | 0.25 | 8 | 0.5 | 1 | 0.125 |
| pMBLe + *bla*KPC-96 Y241N | 0.25 | 0.125 | 2 | 0.5 | 0.5 | 4 | 2 | 2 | 0.125 | 8 | 0.5 | 8 | 0.125 |
| pMBLe + *blaC*KPC-97 V277_I278insNSEAV | 0.125 | 0.125 | 2 | 0.5 | 0.5 | 16 | 4 | 4 | 0.5 | 16 | 1 | 8 | 0.125 |

[a]AVI, avibactam; REL, relebactam.

## DISCUSSION

In this study, we described the emergence of four ceftazidime-avibactam-resistant *K. pneumoniae* isolates carrying novel KPC variants derived from KPC-2. The clinical isolates exhibited uniform resistance to extended spectrum cephalosporins and monobactams. Among the isolates tested, only one exhibited the typical phenotype of complete susceptibility to carbapenems (M25752-*bla*KPC-81), as frequently seen in KPC variants (34). In contrast, the remaining three isolates displayed a mild level of carbapenem resistance. This resistance may be attributed to the intrinsic hydrolytic activity of the newly identified variants, as indicated by the results obtained when these variants were cloned into a susceptible strain which caused a significant elevation of imipenem MICs to intermediate values. The exception was *bla*KPC-81, where the basal TOP10 imipenem MIC values did not change after cloning. Furthermore, the confirmation of the *per se* putative hydrolytic activity of *bla*KPC-80, *bla*KPC-96 and *bla*KPC-97 came through the restoration of full susceptibility to carbapenems upon the addition of avibactam or relebactam.

The susceptibility to carbapenems, specifically meropenem and imipenem, varied among the isolates: except for KPN-*bla*KPC-97, which showed intermediate susceptibility (MIC: 2 µg/mL) to meropenem, the other three clinical strains were susceptible to

**TABLE 4** Bioinformatics data analysis of *K. pneumoniae* isolates

| Parameter | M25399- *bla*KPC-80 | M25752- *bla*KPC-81 | M25197- *bla*KPC-96 | M25923- *bla*KPC-97 |
|---|---|---|---|---|
| Sequence type (MLST) | ST629 | ST258 | ST258 | ST11 |
| Capsular type | K10 | KL106[a] | KL106[a] | K39 |
| KPC-2 allele mutation | K269_D270insPNK | del_I173 | Y241N | V277_I278insNSEAV |
| Loop | Lys270 (267-276) | Omega (164-179) | Val240 (238-243) | Lys270 (267-276) |
| Ariba Resfinder (ID nucleotide) | *bla*KPC (99.78 ID with *bla*KPC-41) | *bla*KPC (99.66 ID with *bla*KPC-2) | *bla*KPC (99.89 ID with *bla*KPC-2) | *bla*KPC (99.01 ID with *bla*KPC-34) |
| Amrfinder (ID protein) | *bla*KPC (98.34 ID with *bla*KPC-58) | *bla*KPC (99.66 ID with *bla*KPC-2) | *bla*KPC (99.66 ID with *bla*KPC-2) | *bla*KPC (96.43 ID with *bla*KPC-44) |
| Genetic platform of *bla*KPC | Tn*4401a* | Tn*4401a* | Tn*4401b-1* | Tn*4401a* |
| Other determinants of resistance | | | | |
| *OmpK35* | Wild type | Frameshift mutation | Frameshift mutation | Frameshift mutation |
| *OmpK36* | Wild type | Wild type | Wild type | Wild type |
| *OmpK37* | Wild type | Wild type | Wild type | Wild type |
| PBP3 (*ftsI*) | Wild type | Wild type | Wild type | Wild type |
| *bla*PER | No | No | No | No |
| *bla*AmpC | No | No | No | No |

[a]Loci identified and labeled KL106, but the corresponding serotypes remains uncharacterized (non-typeable) (32).

meropenem, while only the KPN-$bla_{KPC-81}$ showed susceptibility to imipenem. When we evaluated the MICs of *E. coli* clones harboring the $bla_{KPC}$ allelic variants, we observed that the clones remained susceptible to carbapenems but behaved as resistant to cephalosporins and aztreonam. Furthermore, when we evaluated the same antibiotics in the presence of avibactam, we observed a decrease in the MICs in all cases except for meropenem, whose MICs remained low even in the absence of DBO inhibitors. The same behavior was observed in *E. coli* expressing $bla_{KPC-96}$ with ceftazidime and ceftazidime-avibactam, where the MICs differed only by one dilution.

Three hotspot mutations associated with ceftazidime-avibactam resistance have been described: the omega-loop (Arg164 to Asp179), the Val240-loop (Thr237 to Thr243), and the Lys270-loop (Ala266-Glu275) (7, 11, 35). Each of these sites appears to exhibit a different tolerance to different types of mutations. The omega-loop exhibits flexibility in accommodating both insertions and deletions, while the Val240 loop primarily tolerates deletions and point mutations. In contrast, the Lys270-loop is primarily amenable to insertions, with a majority of these insertions corresponding to duplicated sequences (7, 11, 35). All the variants described in this work fall within this framework of mutations.

When we closely examined each of the new KPC variants, we found that KPC-80 and KPC-97 exhibit insertions/duplications in the Lys270 loop. Interestingly, this loop, while being the second most mutated one linked to ceftazidime-avibactam resistance among KPC variants in the literature, contains fewer mutations compared with other Class A β-lactamases (7). Notably, the same mutations found in KPC-80, derived from KPC-2 in the form of PNK duplication, were also identified in the KPC-41 mutant, originating from KPC-3. In this latter pair, it is observed that KPC-41 has a higher affinity for ceftazidime and lower inhibition by avibactam compared with KPC-3 (9). The PKN duplication has been linked to an enhanced ability to hydrolyze ceftazidime and reduced inhibition by avibactam when compared with KPC-3. In our investigations involving the $bla_{KPC-80}$ clone, we also observed a parallel increase in ceftazidime MICs (Tables 2 and 3). However, it's noteworthy that in this case, the inhibitory effect of avibactam remained unchanged and effective. To substantiate this phenotypic observation, further *in vitro* evaluation of the kinetic parameters of the KPC-80 enzyme is required.

KPC-96 carries a single-point mutation in the Val240-loop, a crucial loop that contains residues involved in the active site and interacts with residues from the omega-loop, such as Val240, Tyr241 and Thr243 (11). Notably, it was this particular variant, when expressed in *E. coli* TOP10 and subjected to avibactam inhibition that exhibited the weakest response on ceftazidime (Table 3). This is the first report of a clinical isolate harboring the Y241N mutation, which until now has only been obtained *in vitro* in laboratory settings following exposure to ceftazidime-avibactam (36).

Finally, KPC-81 has a deletion in the omega-loop. Mutants with deletions in this site are defined as "specialized in the hydrolysis of ceftazidime" as they have been reported to be susceptible to all beta-lactams except for ceftazidime and ceftazidime-avibactam. While the precise mechanism remains not fully understood, it is believed to involve increased binding of ceftazidime (11). The $bla_{KPC-81}$ isolate described here aligns with these previous findings, exhibiting resistance to cephalosporins and monobactams while maintaining susceptibility to carbapenems upon cloning into *E. coli* TOP10 (Table 3). Whenever there was an increase in MICs, the introduction of avibactam successfully restored them to their original baseline values (Table 3). Similar KPC allelic variants have been reported worldwide derived from KPC-3 or KPC-2. For instance, in France, KPC-28 (deletion Δ242-GT-243 derived from KPC-3, with H273Y) was reported (10), and in New York, KPC-14 (deletion Δ242-GT-243 derived from KPC-2) was reported from a clinical isolate in 2003, prior to the introduction of avibactam to clinical treatment (37). In 2020, KPC-14 and KPC-33 (both featuring D179Y mutation) were detected in a patient previously treated with ceftazidime-avibactam in Italy (38). Unfortunately, the emergence of ceftazidime-avibactam-resistant KPC mutants was accelerated with the introduction of this drug in the clinical practice (39). Still, the KPC enzyme mutation alone does not entirely explain the ceftazidime-avibactam resistance phenotype. It is

known that deficiency in the OmpK35 porin also contributes to an increase in this MICs (40). In this work, we found frame shift mutations in the gene encoding for OmpK35 porin in the clinical isolates carrying KPC-81, KPC-96 and KPC-97, although the overall contribution of impermeability to the phenotype seems negligible here.

Of the four *K. pneumoniae* isolates harboring KPC allelic variants described here, three belonged to CC11, M25752 and M25197 to ST258, and M25923 to ST11 (Table 4). In the ST258 isolates, $bla_{KPC}$ was detected within the Tn*4401a* or b-1 genetic element, while in ST11-M25923, it was associated with Tn*4401a*. Historically in Argentina, the ST258 has been associated to the successful dissemination of $bla_{KPC-2}$ primarily in conjunction with Tn*4401a* and not b1 (12) while ST11 has disseminated $bla_{KPC}$ in the Tn*3*-derived genetic elements bearing non-Tn*4401* structures (41). Finally, it is well stablished that *K. pneumoniae* is an opportunistic microorganism known to causes infection through several virulence factors including surface antigens, fimbriae, capsule, outer membrane proteins and siderophores (42). Two of the isolates, M25399 and M25923, were found to harbor genes that encode for K10 and K39 capsular types, respectively. These capsular types have been very frequently described and well characterized as they are known to confer resistance against the bactericidal activity of antimicrobial peptides, complement, phagocytosis and opsonization (43, 44). The other two isolates, M25752 and M25197, harbored capsular loci that has been identified and labeled as KL106, but the corresponding serotypes resulted as non-typeable because they remain uncharacterized (32).

## Conclusions

In this study, we report the emergence of resistance to ceftazidime-avibactam due to the emergence of allelic variants of $bla_{KPC-2}$. These $bla_{KPC}$ alleles harbored mutations that were in hotspots of the mature protein. This underscores the importance of both epidemiological surveillance and molecular investigations in monitoring and comprehending the diverse mechanisms of resistance that pose challenges to effective antimicrobial treatments. Moreover, it is essential to exercise caution and judiciously use of ceftazidime-avibactam in the clinical practice in order to prevent the emergence of resistance. Preserving the effectiveness of this antibiotic is crucial to curb the spread of resistance.

## ACKNOWLEDGMENTS

This study is supported by the Agencia Nacional de Promoción Científica y Tecnológica (ANPCyT), PICT-2017-0321 to S.A.G. and the Ministry of Health, annual budget assigned to Antimicrobial Division (NRRLAR). S.A.G. is member of Carrera del Investigador Científico, CONICET, Argentina. M.B.S. is a research fellow of ANPCyT, under the PhD scholarship program through grant PICT-2017-0321.

We would like to thank Stella Cristaldo for the technical assistance and Dr. Denise de Belder, Federico Lorenzo, and Dr. Josefina Campos for the short-read sequencing at the Unidad Operativa Centro Nacional de Genómica y Bioinformática, ANLIS "Dr. Carlos G. Malbrán," Buenos Aires, Argentina.

## AUTHOR AFFILIATIONS

[1]National and Regional Reference Laboratory in Antimicrobial Resistance (NRRLAR)-INEI-ANLIS Dr. Carlos G. Malbrán, Buenos Aires, Argentina

[2]Universidad de Buenos Aires, Facultad de Farmacia y Bioquímica, Instituto de Investigaciones en Bacteriología y Virología Molecular (IBaViM), Buenos Aires, Argentina

[3]Consejo Nacional de Investigaciones Científicas y Técnicas (CONICET), Buenos Aires, Argentina

[4]Hospital J.A. Fernández, Buenos Aires, Argentina

[5]Hospital Provincial Neuquén Dr. Castro Rendón, Neuquén, Argentina

[6]Hospital Británico, Buenos Aires, Argentina

[7]Laboratorio Stamboulián, Buenos Aires, Argentina

## AUTHOR ORCIDs

María Belén Sanz ⓘ http://orcid.org/0000-0002-6087-0388
Pablo Power ⓘ https://orcid.org/0000-0002-7051-9954
Sonia A. Gomez ⓘ http://orcid.org/0000-0001-9106-1762

## FUNDING

| Funder | Grant(s) | Author(s) |
|---|---|---|
| MINCyT \| Agencia Nacional de Promoción Científica y Tecnológica (ANPCyT) | PICT-2017-0321 | Sonia A. Gomez |

## AUTHOR CONTRIBUTIONS

Sonia A. Gomez, Conceptualization, Data curation, Formal analysis, Funding acquisition, Investigation, Project administration, Supervision, Writing – original draft, Writing – review and editing.

## DATA AVAILABILITY

The sequenced data have been deposited in NCBI under the BioProject PRJNA996143. Accession numbers are as follows: *K. pneumoniae* M25923, GenBank no. JAVKZB000000000.1 and BioSample no. SAMN36519330; *K. pneumoniae* M25197, GenBank no. JAVKZC000000000.1 and BioSample no. SAMN36519320; *K. pneumoniae* M25752, GenBank no. JAVKZD000000000 and Biosample no. SAMN36519319; *K. pneumoniae* M25399, GenBank no. JAVKZE000000000 and Biosample no. SAMN36518399. The allelic variant accession numbers are MW444845.1 (KPC-80), OK086970.1 (KPC-96), OK086971.1 (KPC-97), and MW444846.1 (KPC-81).

## ADDITIONAL FILES

The following material is available online.

### Supplemental Material

**Supplemental tables (Spectrum04111-23-S0001.xlsx).** Table S1 to S3.

### Open Peer Review

**PEER REVIEW HISTORY (review-history.pdf).** An accounting of the reviewer comments and feedback.

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
