## [Reviewer comments · Microbiology Spectrum]

Microbiology Spectrum

KPC-2 allelic variants in *Klebsiella pneumoniae* isolates resistant to ceftazidime-avibactam from Argentina: bla_{KPC}-80, bla_{KPC}-81, bla_{KPC}-96 and bla_{KPC}-97

Maria Sanz, Fernando Pasteran, Juan De Mendieta, Florencia Brunetti, Ezequiel Albornoz, Melina Rapoport, Celeste Lucero, Laura Errecalde, Maria Nuñez, Renata Monge, Magdalena Pennini, Pablo Power, Alejandra Corso, and Sonia Gomez

Corresponding Author(s): Sonia Gomez, Servicio Antimicrobianos, Departamento de Bacteriología, Instituto Nacional de Enfermedades Infecciosas - ANLIS

Review Timeline:

Submission Date:

December 11, 2023

Accepted:

January 10, 2024

Editor: Krisztina Papp-Wallace

Reviewer(s): The reviewers have opted to remain anonymous.

Transaction Report:

DOI: <https://doi.org/10.1128/spectrum.04111-23>

Re: Spectrum04111-23 (KPC-2 allelic variants in *Klebsiella pneumoniae* isolates resistant to ceftazidime-avibactam from Argentina: bla_{KPC-80}, bla_{KPC-81}, bla_{KPC-96} and bla_{KPC-97})

Dear Dr. Sonia A Gomez:

Your manuscript has been accepted, and I am forwarding it to the ASM production staff for publication. Your paper will first be checked to make sure all elements meet the technical requirements. ASM staff will contact you if anything needs to be revised before copyediting and production can begin. Otherwise, you will be notified when your proofs are ready to be viewed.

Sincerely,
Krisztina Papp-Wallace
Editor
Microbiology Spectrum

Reviewer #1 (Comments for the Author):

In this work, authors characterized by phenotypic and genotypic methods 4 Kp strains producing KPC-2 variants that confer resistance to ceftazidime-avibactam. The work is well-written, but it has a relatively low impact. In fact, there are many papers in this context and the present one does not offer kinetic data (that are essential in my opinion).

Major comments

1. L55. OmpF???? This is not clear for me. OmpF is in *E. coli*, *Enterobacter*, or *Salmonella*. In addition, I do not see OmpF in Table 4. Please clarify.
2. Results section is too long. Your Tables are enough detailed. Please cut down text.
3. Clinical data. Do you know if the 4 patients had a previous infection with a KPC-producing Kp or other Gram-negatives? This is an important information, especially because 2 patients received a previous treatment with caz-avibactam.
4. Authors indicated that the 4 variants of bla_{KPC} were always in Tn4401-like and also that one strain was conjugative. However, there is a complete lack of genetic environment description. Which plasmid(s)? Were the remaining Tn4401-like chromosomally located?
5. Discussion section is redundant with Introduction and Results. In addition, authors should include a discussion about the lack

of kinetic data.

Minor comments

6. L119-123. Which company provided disks?

7. L129-134. Authors should provide specific reference for CLSI and EUCAST.

8. L144-146. Any reference for the primers used to detect OmpK35-36?

9. L168-170. How recombinant plasmids were sequenced?

10. In my opinion, Tab 2 and Tab 3 can be merged.